# Correspondence between the Compositional and Aromatic Diversity of Leaf and Fruit Essential Oils and the Pomological Diversity of 43 Sweet Oranges (*Citrus x aurantium* var *sinensis* L.)

**DOI:** 10.3390/plants12050990

**Published:** 2023-02-21

**Authors:** Vincent Ferrer, Noémie Paymal, Gilles Costantino, Mathieu Paoli, Carole Quinton, Félix Tomi, François Luro

**Affiliations:** 1INRAE, UMR AGAP Institut, INRAE, Institut Agro, Cirad, University Montpellier, 20230 San Giuliano, France; 2Rémy Cointreau–Les Molières, 49124 Saint-Barthélemy-d’Anjou, France; 3CNRS, Equipe Chimie et Biomasse, UMR SPE 6134, Université de Corse, 20000 Ajaccio, France

**Keywords:** GC (FID), GC/MS, SSR, CATA, yield of essential oil, fruit shape, pulp acidity, TSS, CCI

## Abstract

Orange (*Citrus x aurantium* var *sinensis*) is the most widely consumed citrus fruit, and its essential oil, which is made from the peel, is the most widely used in the food, perfume, and cosmetics industries. This citrus fruit is an interspecific hybrid that would have appeared long before our era and would result from two natural crosses between mandarin and pummelo hybrids. This single initial genotype was multiplied by apomictic reproduction and diversified by mutations to produce hundreds of cultivars selected by men essentially based on phenotypic characteristics of appearance, spread of maturity, and taste. Our study aimed to assess the diversity of essential oil composition and variability in the aroma profile of 43 orange cultivars representing all morphotypes. In agreement with the mutation-based evolution of orange trees, the genetic variability tested with 10 SSR genetic markers was null. The oils from peels and leaves extracted by hydrodistillation were analyzed for composition by GC (FID) and GC/MS and for aroma profile by the CATA (Check All That Apply) method by panelists. Oil yield varied between varieties by a factor of 3 for PEO and a factor of 14 for LEO between maximum and minimum. The composition of the oils was very similar between cultivars and was mainly dominated by limonene (>90%). However, small variations were observed as well as in the aromatic profile, with some varieties clearly distinguishing themselves from the others. This low chemical diversity contrasts with the pomological diversity, suggesting that aromatic variability has never been a selection criterion in orange trees.

## 1. Introduction

The sweet orange (*Citrus x aurantium* var *sinensis*, or *C. sinensis* L.) is the fifth most consumed fruit in the world, and global production is estimated to be approximately 70 million tons [1]. Representing two-thirds of consumption, the fruit is mainly consumed fresh, with the remaining third being processed into juice [2]. This juice industry generates many coproducts, mainly orange peel and essential oils [3,4]. The global production of orange essential oil is estimated to be approximately 60,000 tons, making it a major product for flavoring beverages, food products, and cosmetics [5]. The chemical composition of orange essential oil has been studied many times, and several bibliographic reviews are available [6,7]. The composition is totally dominated by monoterpenes (mainly limonene), but there are also some sesquiterpenes in small proportions as well as aliphatic aldehydes. The aromatic properties of the different compounds in orange peel were studied by gas chromatography coupled with olfactometry [8,9,10,11,12,13,14]. This shows that aliphatic aldehydes (octanal, nonanal, decanal, and dodecanal) and oxygenated monoterpenes (linalool, citronellal, and citral) are the major contributing compounds to orange aroma, whereas hydrocarbon monoterpenes (limonene, myrcene, and α-pinene) and sesquiterpenes (mainly β-sinensal) have a small contribution. Nevertheless, the PEO composition of a citrus fruit can be influenced by many factors [15], such as the environment [16], the harvest year [17], the harvest date [18,19], the rootstock [20,21], the extraction system [22,23], the storage conditions [24], and the health of the tree [25,26,27]. Some examples of the effects of these different factors of variation in citrus PEO composition are presented hereafter. The environments of Bahia (Brazil) and Corsica (France) modified the proportions of limonene, neral, geranial, and derivatives in citron and lemon PEOs [16]. Gioffrè et al. reported the chemical composition of the bergamot peel essential oil described in the studies carried out from 1984–1985 until 2018–2019 and highlighted noticeable variations, with the major component limonene ranging between 31.10% and 44.48% [17]. The PEO composition was richer in oxygenated compounds at early fruit development stages, with an aromatic profile presenting greener notes than in mature orange fruit [19]. The rootstock influenced the yield and a little bit the composition of the orange PEO, but with a very low impact on flavor [20]. The citrus EO extracted by microwave—the Clevenger method—contained elevated quantities of oxygenated compounds compared to hydro-distillation and cold pressing methods [22]. In sour orange, linalyl acetate and linalool drastically decrease when citrus are ripening [10]. The longer the oil was stored, the more important the changes in the composition were: limonene, myrcene, γ-terpinene and linalyl acetate percentages decreased when the temperature and the time of storage increased [24]. The PEOs of orange trees infected by Huanglongbing (HLB) had lower concentrations of valencene, octanal, and decanal and were abundant in oxidative/dehydrogenated terpenes, such as carvone and limonene oxides [27].

The birth of the orange tree is thought to date back several millennia to China and is the product of a cross between two mandarin/pummelo hybrids [28]. This initial hybrid would have multiplied by somatic embryony, been diversified by apomixis-fixed mutations, and then been propagated by cuttings and grafting when discovered by humans. As a result, genetic diversity among cultivars is very low [28,29,30,31,32]. There are a large number of varieties with diverse tastes, shapes, and colors that have been selected by humans [33,34,35,36]. These cultivars have been classified into four phenotypic groups: ‘blond,’ ‘blood,’ ‘acidless,’ and ‘navel’. The sanguine cultivars are the result of a natural genetic modification related to the displacement and insertion of a transposon element in the promoter sequence of the *Ruby* gene, which causes anthocyanin synthesis [37]. The first navel orange appeared in Brazil (Bahia state) in 1820, following a spontaneous mutation that made an atrophied embryo of a second orange appear on the higher part of the fruit. The stylar scar took the form of a growth reminiscent of a navel. This new orange was introduced in the USA in 1870 and took the name of ‘*Washington navel*’. All the cultivars of navel oranges selected thereafter were derived from this ‘Washington navel.’ Acidless oranges are cultivars derived from acid pulp cultivars whose mutations have caused the juice to become much less acidic.

Significant variation in zest essential oil yield has been observed between varieties of ‘blond’ oranges. The cultivar ‘Valencia’ produces on average twice as much oil as the cultivar ‘Hamlin’ [38]. The most robust studies on the subject identified significant variations in oxygenate content between the different cultivars, with notably higher levels in ‘blond’ cultivars [39,40,41]. With nearly 150 accessions, the INRAE-CIRAD citrus collection at San Giuliano (France) presents a significant panel of orange diversity in the form of trees grown in orchards under the same conditions. These genetic resources constitute an adequate device to evaluate within the group the diversity of the composition of orange essential oils and their aromatic profiles.

## 2. Results

### 2.1. Genetic Diversity of Oranges

The ten SSR markers revealed no polymorphisms among the 43 orange cultivars.

### 2.2. Pomological Diversity of Oranges

The different fruit descriptors are variable (Figure 1 and Appendix A). Regardless of the cultivar, the shape of the fruit was quite close to a round shape (DP/DE between 0.9 and 1.1), except for the ‘*Pera*’ and ‘*Shamouti’ cultivars*, which had an ovate shape (DP/DE > 1.2). Finally, the color of the peel was close to orange (CCI approximately 5) for most cultivars except for ‘Huan Pi Chen,’ which was yellow (CCI close to 0), and ‘*Navelina*’ had a deep orange color (CCI > 8). The acidity of the fruit pulp was also slightly variable except for the cultivars without acidity, ‘*Sakkaria lokum*’ and ‘*Iaffaoui douce*’, and two cultivars, ‘*Natal*’ and ‘*Lue Gim Gong*,’ with very acidic pulp (>1.6%), while the average acidity was 0.9%. The most variable parameters are fruit mass (150 to 420 g), skin thickness (3 to 8 mm), soluble sugar content (8.5 to 13° Brix), and essential oil yield (2.8 to 9.1% with an average value of 5.8%).

Clustering of oranges based on phenotypic variables, including essential oil yields, discriminates the cultivars essentially into two distinct lots (‘*Sanguinelli*’ to ‘*Iaffaoui douce*’) and (‘*Natal*’ to ‘*Portuguaise*’) (Figure 2). This differentiation is based mainly on two characteristics: the thickness of the skin and the mass of the fruit, which are generally higher in the first group. In the second group, two substructures emerge: that of oranges with low TSS, skin thickness, and fruit mass, i.e., from ‘*Parson-Brown*’ to ‘*Portugaise*’ and that of grouping cultivars from ‘*Cara Cara Navel*’ to ‘*Salustiana*,’ which are distinguished by high sugar content (TSS). In this last group of seven cultivars, four are blood oranges. Indeed, the average weight of a fruit for all the ‘non navel’ cultivars is approximately 220 g, whereas for the ‘navel’ oranges, it is 375 g. The ‘navel’ oranges also seem to be slightly less acidic, more elongated, and have a rind that is on average one centimeter thicker than the other cultivars.

The ‘blond’ and ‘blood’ oranges have similar overall compositions, with soluble sugar contents that appear to be higher than those of ‘navel’ oranges. Other characteristics, such as the number of segments and the oil yield, seem similar between the different groups of varieties.

The oil yield of the leaves had a high variation amplitude between 0.001 and 0.137% for an average value of 0.062% (Appendix A). The most productive cultivar was ‘*Huan Pi Chen*,’ while the five worst yielding cultivars were ‘blood’ oranges.

### 2.3. Diversity of LEO Composition

Forty-eight compounds were detected and identified among the 43 orange cultivars, representing between 97.2 and 99.7% of the total (Appendix A). For PEO, monoterpenes are the most important, representing between 91.3 and 97.9% of the total. The main monoterpene is sabinene, with an average value of 41.5% and varying between 10.4% and 41.5%. The proportion of oxygenated monoterpenes is high, at 27.2% on average, and varies between 9.5 and 48.6%. Linalool is the main alcohol, which account approximately for 9.3% of the total compounds. There are also aldehydes, such as citronellal and citral (neral/geranial), at approximately 2.0%, as well as acetates, such as geranyl and neryl acetate, all below 1%.

With an average value of 3.2% and a proportion varying between 0.8 and 5.9%, the sesquiterpenes were the second dominant family in the LEO. The main compound was β-sinensal (approximately 2%). The α-sinensal and the β-elemene were approximately 0.4%, and the other compounds were in very low proportions or even at trace levels. Among the aliphatic compounds (0.3–1.2%), 2-hexenal is the main compound and varies between 0.2 and 1.0% (on average 0.5%). Decanal, hexanal, and hexanol were found in trace amounts or in very low proportions. Finally, diterpene was identified, (E)-phytol, with an average concentration of 0.2% but sometimes detected only at trace levels and up to 0.6%.

The only difference in families of compounds between the two types of orange trees was observed for monoterpene aldehydes, which exhibit an average content more important in ‘navel’ oranges than in ‘blood’ oranges (Table 1).

The diversity of cultivars represented by the heatmap based on the 23 major compounds of LEO was structured into three clusters (Figure 3). One of these clusters included 23 cultivars (from ‘*Natal*’ to ‘*Navelina SG*’), another 5 cultivars (from ‘*Portugaise*’ to ‘*Salustiana*’), and the third one included 15 cultivars (from ‘*Shamouti*’ to ‘*Parson Brown*’). Except for the second cluster of 5 cultivars grouped by a chemical profile characterized by low amounts of (E)-β-ocimene, γ-terpinene, α-terpinene, terpinolene, and β-phellandrene, no chemical signature exists for the two other clusters. However, two triplets of cultivars stood out from other oranges. The first one (‘*Fukhuara*’, ‘*Pineapple*’, and ‘*Navelina SG*’) was characterized by high contents of geranial, neral, and citronellal, and the second one (‘Salustiana’, ‘*Tarocco Rosso*’, and ‘*Portugaise*’) was characterized by low contents of (E)-β-ocymene, γ-terpinene, α-terpinene, terpinolene and β-phellandrene.

### 2.4. Compositional Diversity of Peel Oil Diversity of PEO Composition

Fifty-nine compounds were identified among the 43 orange cultivars studied, representing between 99.99% and 100.00% of the total oil composition (Appendix A). Monoterpenes were the ultra-majority, representing between 98.8% and 99.7% of the composition. Among the hydrocarbon monoterpenes, the limonene percentage ranged from 92.4% to 96.1%. Myrcene was approximately 2%, sabinene close to 1% and α-pinene at approximately 0.5%. The majority of other hydrocarbon monoterpenes were present at trace levels. Oxygenated monoterpenes showed a significant variability between 0.4% and 1.8% for an average value of 1.1%. Linalool was the main compound (approximately 0.70%) and was responsible for the variation in the content of oxygenated monoterpenes because it varies between 0.2% and 1.4%. Other compounds were found in very low proportions, such as terpinen-4-ol, α-terpineol, citronellal and citral (neral/geranial). Other compounds were detected only at trace levels.

Aliphatic compounds represented between 0.2% and 1.2% of the composition of the oil, with an average of 0.54%, and were mainly aldehydes. Nevertheless, traces of alcohol and esters were detected. The two main compounds, octanal and decanal, varied between 0.1–0.6% and 0.1–0.5%, respectively. The other aldehydes detected were nonanal, hexanal, 2-hexenal, and dodecanal, all in trace amounts. Finally, the sesquiterpenes represented approximately 0.2%. The main compound was valencene, accounting for approximately 0.1%. Two aldehydes, α- and β-sinensals, and several hydrocarbon sesquiterpenes, δ-cadinene, α-farnesene, and (E)-β-caryophyllene, were also present at trace levels.

It appears that the aliphatic aldehyde content is higher in ‘navel’ oranges than in ‘blood’ oranges (Table 2). ‘Blood’ oranges also appear to be lower in monoterpene aldehydes than ‘blond’ oranges. The other types of compounds had equivalent contents between the cultivars.

Four clusters can be distinguished in the diversity of the cultivars analyzed by the 20 majority compounds of the peel essential oil (Figure 4). One of them, at the top of the heatmap, includes 9 cultivars (‘*Portugaise*’ to ‘*Sanguinello Moscatata Cuscuna*’), characterized by high levels of valencene, α-pinene, β-pinene, sabinene, γ-terpinene, and terpinene-4-ol. Among these 9 cultivars, 5 are ‘blood’ oranges. A subgroup of 7 cultivars (‘*Cam Nat*’, ‘*Tarroco Rosso*’, ‘*Sanguinelli*’, ‘*Pera*’, ‘*Boukhobza*’, ‘*Sakkaria Lokum*’ and ‘*Tarocco*’) were distinguished from others by having higher than average contents of β-phellandrene and limonene and conversely lower contents of β-sinensal and α-sinensal. The highest levels of decanal, octanal, and dodecanal characterized the chemical profiles of ‘*Navelina*’ and ‘*Navelina SG*’. No chemical signature was observable for the last two groups.

### 2.5. Aromatic Diversity of the Peel Oil

A first set of triangular tests was conducted between four samples of phenotypically different orange peel essential oils to assess the existence of aromatic diversity (Figure 5 and Appendix A). Of the six combinations, five indicated that the oils were perceived as different between cultivars, and only ‘*Valencia Late*’ and ‘*Shamouti*’ showed confounding profiles by the panelists.

To refine the understanding of this diversity, expert panelists from Rémy-Cointreau used the CATA (Check All That Apply) method to analyze essential oils of all varieties. Of the 29 descriptors used, 15 were significantly different between varieties according to Cochran’s test (*p* ≤ 0.05): ‘Sweet Orange’, ‘Orange Juice’, ‘Zesty’, ‘Candied’, ‘Lavender’, ‘Pepper’, ‘Cardamom’, ‘Nutmeg’, ‘Lemon Grass’, ‘Lemon’, ‘Blood Orange’, ‘Oxidized’, ‘Hydrocarbon’, ‘Alembic Juice’ and ‘Fat’.

A correspondence factor analysis on the contingency table of the CATA test results highlights the diversity of orange flavor profiles (Figure 6 and Appendix A). The Pearson chi-square test (*p* = 0.002) performed on the 15 descriptors mentioned above from this same contingency table confirms that the cultivars are different for these descriptors. Varietal diversity according to aroma profile was revealed by a spread of the varietal cloud on the first axis of the CFA, representing 23% of the total diversity. On this first axis, the cultivars ‘*Pera*’, the two ‘*Maltaise demi-sanguine*’ and ‘*Fisher navel*’ clearly stand out from the group with notes of ‘nutmeg’, ‘cardamom’, ‘jus-alambic’ and ‘oxide’. Like these cultivars, this axis differentiates ‘*Bisri*’, ‘*Sanguinello*’, ‘*Navelate*’ and ‘*Valencia late*’ from the other sweet oranges. On the second axis of the CFA (13% of the total variation), two varieties emerge from the group, ‘*Parson Brown*’ (‘pepper’ note) and ‘*Hamlin*’ (‘bold’ and ‘lavender’ notes).

## 3. Discussion

The lack of genetic diversity observed with SSR markers is consistent with the evolutionary hypothesis for this citrus group that it diversified through mutations and clonal selection [28,29,30,31,41]. Varietal selection has been performed in orchards mainly on easily observable visual or taste criteria, such as sugar content, acidity, color, fruit size, and number of seeds [42]. This explains the observed phenotypic diversity within this group [32,33,34,35,36,43,44]. The classification into four phenotypic groups (‘navel,’ ‘blood,’ ‘blond,’ and ‘acidless’) does not appear in the overall diversity based on phenotypic traits. It is probable that if the characters ‘navel’ or ‘blood color of the pulp’ were added, the structure would have been different.

The yield of PEO is variable between oranges. However, our results do not coincide with those observed by Kesterson and Braddock (1975) on four cultivars common to the sample: ‘*Hamlin*,’ ‘*Parson Brown*’, ‘*Pineapple*,’ and ‘*Valencia*’ [37]. This study was conducted in Florida with orange trees grafted on a different rootstock than the one used in Corsica (rough lemon versus Carrizo citrange). In Kesterson and Braddock’s study, the ‘*Valencia*’ cultivar had a higher yield (13.5 lb/ton of fruits), and ‘*Hamlin*’ had a lower yield (7.8 lb/ton of fruits). Among the same four cultivars in the study made in Corsica, ‘*Parson Brown*’ gave the highest yield (9.1%, g/100 g), and ‘*Pineapple*’ gave the lowest yield (7.8%). The method of calculation of the yield was different in the two studies; thus, it is not possible to compare the values. The LEO yield is generally 1.5 to 2 times lower than what is observed in the literature [45,46,47,48]. However, in most publications, the LEO yield value is given in *v*/*w*, whereas in our study, it is expressed in g/100 g of leaves. Despite the correction of the value by considering the density (0.85), the yields of our study remain lower than those in the publications. The explanation could come from the quantity of leaves. The differences could be explained by the loss of a partial quantity of essential oils in the Clevenger piping at the time of their recovery, a loss all the more impactful as the quantity of oil to be recovered is low. In other studies, the quantity of leaves ranged from 250 g to 800 g and was only 100 g in our study. No details are given in these articles on the method of measuring the volume of EO; it can only be assumed that, given the accuracy of the provided value (to one decimal point), it was evaluated on the graduated column of the Clevenger where the oil is recovered. Apart from the values, the differences in yield between varieties are sometimes comparable; for example, the cultivar ‘*Valencia*’ produces 1.6 times more oil than ‘*Washington navel*’ and 1.3 times more oil in our study [45]. Furthermore, it seems that thicker skin cultivars have a lower yield than thinner skin cultivars, although this is calculated with respect to the dry mass of the skin.

The overall composition of the peel and leaf oils agrees with the literature [6,7]. There are no specific compounds related to morphological groups (‘blood,’ ‘navel,’ or ‘blond’) for either peel or leaf oils. This suggests that the varietal selection was made only on the basis of the visual characteristics of the fruit and the composition of the juice, and that the chemical variability of the essential oil is mostly incidental. In the leaf oils, significant differences were observed between large groups of cultivars in monoterpene aldehyde content and in the zest in aliphatic and monoterpene aldehyde content. ‘Blood’ cultivars have already been identified as having lower levels of these compounds [38,39,40]. These aldehyde compounds are known to be major contributors to orange peel aroma [8,10,12,13]. These differences between phenotypic groups would suggest that the cultivar initially selected on a pomological trait other than essential oils may have also simultaneously undergone genetic changes in its biosynthetic pathway for these families of compounds, changes that were then passed on to subsequent selections made from the initial cultivar.

Even relatively small amounts of flavor profile diversity among orange peels have been established. Gaffney et al. (1996) had previously observed that the flavor profiles of zests from ‘blond’ orange cultivars were close enough to be interchangeable [9]. This result is unsurprising considering that the diversity of this group is based solely on clonal selection of spontaneous mutations.

There was no correlation between the variation in composition and the variation in aromatic profile. This is probably due to several reasons. First, a number of compounds are at levels below the GC detection limit and remain undetectable, but they probably contribute to the differentiation in aroma profiles [49]. It is likely that these minority compounds explain the aromatic differences between ‘blond’ and ‘blood’ orange juices and between peel oils of different cultivars [32]. The presence of limonene in very large quantities, which certainly has only a minor aromatic role, masks the detection of minority compounds by GC [50,51,52]. Finally, the CATA method is probably not suitable for measuring such small differences. It would seem wise to perform GC-O analyses to understand whether the observed differences are simply due to changes in compound proportions or to the existence of cultivar-specific compounds. It might also be interesting to perform fractionation to remove limonene via molecular distillation or chromatography, which would increase the concentrations of minority oxygenates.

However, some cultivars present aromatic profiles very different from the majority of the group. An analysis focused on these varieties is therefore planned and will be carried out with biological replicates and with a method facilitating the detection of minor compounds and the study of their quantitative variations. The reduction of the PEO samples will also allow a better description of their aromatic profiles.

## 4. Materials and Methods

### 4.1. Biological Materials

All the biological materials used in this study came from the orchard of the INRAE-CIRAD citrus collection at San Giuliano, France (latitude 42°17′ N, longitude 9°32′ E; Mediterranean climate; average rainfall of 840 mm per year; average temperature of 15.2 °C; soil derived from alluvial deposits and classified as fersiallitic, with a pH range of 5.0–5.6) [53]. The average annual temperature of the year from blossom (May 2018) to leaf picking (June 2019) was 15.2 °C, with a mean maximum of 30.3 °C in July and a mean lower of 3.6 °C in January. The mean temperature of day and night difference during this period was 11.7 ± 2.9 °C. Forty-three orange cultivars grafted on the same rootstock (Carrizo citrange, [*C. sinensis* × *Poncirus trifoliata*]) and grown under identical conditions were used for the analysis of phenotypic and aromatic diversity. The planting density is 416 trees per hectare. Fertilization was provided in three annual applications (February, May, and August) for a total of 450 kg per hectare with *Multigro* fertilizer (Haifa, France), 15% nitrogen, 23% phosphoric anhydride, 14% potassium oxide, and 2% magnesium oxide. The trees were irrigated by micro-sprinklers according to the estimated needs of the tree with respect to the potential evapotranspiration (Etp), rainfall (P), and cultural coefficient (Kc): (Etp–P) × Kc. The water supply during the dry summer period was approximately 25 mm per week, for a total of about 1000 L per tree per year.

The 43 accessions (Appendix A) were selected from the 150 cultivars in the collection to have a representation of the different morphotypes.

### 4.2. Phenotypic Description

#### 4.2.1. Data Acquisition

To perform the phenotypic analysis, five representative fruits from three different trees, each with forty-three accessions of sweet orange, were harvested during the first week of March 2019 (300 to 315 days after blossom). In this study, all the fruits were individually measured.

The polar and equatorial diameters (PD and ED) of each fruit were measured using a caliper model IP 67 (BLET, Rueil-Malmaison, France), and their ratio (PD/ED) yields the shape of the fruits: flattened for values below one and oblong for values higher than one. The weight (W) of each fruit was measured using a balance model, Pionner (Ohaus, Parsippany, NJ, USA). The thickness of the peel (PT) was measured using a caliper IP 67 model (BLET, Rueil-Malmaison, France). The number of segments (NS) of each fruit was individually counted. The color of the flavedo fruit peel was measured using a colorimeter model Chroma meter CR-400 (Konica Minolta Sensing, Ramsey, NJ, USA) with the determination of a*, b*, and L color indices, where a* corresponds to the variation between green and red, b* corresponds to the variation between blue and yellow, L is the brightness variation between black and white. The combination of these variables permits calculating the Citrus Color Index (CCI). These variables were then applied in the formula CCI = 1000 × a*/(L × b*), with variation between −20 (green) and +20 (orange), where zero (0) corresponded to yellow [54]. Each fruit was measured four times around the equatorial line.

Five fruits from each tree were separately hand-pressed, and equal proportions were mixed in three lots per cultivar to perform the following juice analysis. The total soluble solid (TSS) content expressed in Brix was measured three times using an RFM710 refractometer (Bellingham+ Stanley^®^, Weilheim in Oberbayern, Germany). The acidity (AC) expressed in percent (gram of citric acid per 100 g of juice) was measured three times using an 855 Robotic Titrosampler, (ΩMetrohm^®^, Herisau, Switzerland).

#### 4.2.2. Statistical Analysis

The distribution of each parameter (W, CCI, NS, PT, TSS, AC, and PDED) was represented using boxplots using the R (v4.0.1) package ‘ggplot2′ [55]. Significant differences (*p* ≤ 0.05) within cultivars were calculated for each parameter using one-way analysis of variance (ANOVA) and Tukey’s test using the ‘agricolae’ package in R [56]. Heatmaps representing the varietal diversity according to pomological parameters and the oil composition of peels and leaves were made with the basic package of the Rstudio software (v4.0.1) [57]. The data were previously centered and reduced.

### 4.3. Essential Oil Analysis

#### 4.3.1. Raw Materials

To perform the PEO extraction, the fruits were harvested in January 2019 and hand-peeled in order to obtain approximately 250 g of fresh material for the 43 sweet orange accessions. The fruits were picked randomly from three different trees.

The LEO extraction was conducted with 200 g of leaves at their maximum development point that were harvested in June 2019 from three different trees of the forty-three cultivars.

#### 4.3.2. Hydrodistillation

Fruit peel (250 g) was blended with distilled water for 1 min using a blender model 1300 W (Magimix, Vincennes, France). The leaf material was not blended.

The samples were reacted in a 2 L wide neck flask with a final volume of 1 L (sample and distilled water) and heated for 2.5 h using a heating mantle model EM2000/CE (Electrothermal^®^, London, UK) thermostat 7. The essential oil was collected using a classical Clevenger apparatus. The Clevenger apparatus was cooled using a refrigerated fluid (a mix of glycol/water) cooled to 4 °C and moved by a mini-chiller model C20 (Huber^®^, Offenburg, Germany).

Then, the essential oils were stored in an overfilled 300 µL tainted vial and stored at −20 °C before further analysis. The zest oil yield was calculated from the dry mass of zest measured from a portion of the sample that was oven-dried (50 °C, 24 h). The oil yield of the leaves was calculated from the fresh mass of leaves.

#### 4.3.3. Essential Oil Analysis Using Gas Chromatography and Gas Chromatography-Mass Spectrometry

Gas chromatography analyses were performed on a Clarus 500 gas chromatograph (PerkinElmer, Waltham, MA, USA) equipped with a flame ionization detector and equipped with two fused silica gel capillary columns (50 m length, 0.22 mm i.d., film thickness of 0.25 μm), BP-1 (polydimethylsiloxane) and BP-20 (polyethylene glycol). The oven temperature was programmed to rise from 60 to 220 °C at 2 °C/min and then held isothermal at 220 °C for 20 min, with injector temperature 250 °C, detector temperature 250 °C, carrier gas hydrogen (1.0 mL/min), and split 1/60. The relative proportions of the oil constituents were expressed as percentages obtained by peak area normalization without using correcting factors. Retention indices were determined in relation to the retention times of a series of *n*-alkanes (C7–C28) with linear interpolation (“Target Compounds” software of Perkin Elmer).

Gas chromatography coupled with mass spectrometry was conducted with a TurboMass quadrupole detector (Perkin Elmer, Waltham, MA, USA) directly coupled to an Autosystem XL (Perkin Elmer), equipped with a fused silica gel capillary column (50 m length, 0.22 mm i.d., film thickness of 0.25 μm), and BP-1 polydimethylsiloxane). Carrier gas, helium, at 0.8 mL/min; split 1/75; injection volume, 0.5 μL; injector temperature, 250 °C; energy ionization, 70 eV; electron ionization mass spectra were acquired over the mass range 40–400 Da. The identification of components was based: on comparison of their gas chromatography retention indices on polar and apolar columns, determined relative to the retention times of a series of n-alkanes with linear interpolation with those of authentic compounds and literature data; on computer matching against the National Institute of Standards and Technology (NIST) commercial mass spectral library; and on comparison of spectra with literature data. For further information, refer to the publication by Luro et al. [58].

#### 4.3.4. Statistical Analysis

A one-factor analysis of variance (ANOVA) was performed to test the likelihood of the observed differences between cultivar groups based on oxygenate contents. Statistical groups were formed using Tukey’s test with the ‘agricolae’ package in R (v4.0.1) [56].

The structure of all the LEO cultivars was graphically represented using a heatmap. The analysis was performed using the basic package plots in R [57]. The data were previously centered and reduced.

### 4.4. Sensorial Analysis of Sweet Orange Peel Oil

#### 4.4.1. Tests, Panelists, and Descriptors

The panelists (ten women and eight men aged between 22 and 55) were not smokers; they were people from the Cointreau laboratory trained to control the quality of the liquors during the manufacturing process and participate in daily tastings.

The essential oils of four cultivars were compared via triangular tests on a panel of 18 experienced panelists. The oils were presented in stained tubes described by three-digit numbers, making them indistinguishable. The panelists had to indicate the different tube among the three presented, taking the necessary time to smell the samples.

A “check all that apply” (CATA) questionnaire was used. Twenty-nine descriptors were selected during consensus training sessions of the Cointreau Expert Panel, from a few randomly selected samples of bitter orange essential oils. These descriptors are terms that the Expert Panel already uses for Cointreau distillates already identified during CG-O sessions on the distillate.

The pure, distilled PEO sample was presented in tainted vials to the panelists. Each PEO sample was identified by a code, making it unidentifiable by panelists, and given in a random order without prior information. The panelist had one minute to smell the sample using a test strip and tick off the perceived aromatic characteristics in the provided chart. The question asked to the panelists is the following: “Which of the twenty-nine descriptors characterizes the aroma of the sample?” Five distinct sessions were conducted for the analysis of the forty-three samples.

#### 4.4.2. Statistical Analysis

The statistics of the triangular tests were carried out with the function discrimination of the package ‘sensR’ of the software Rstudio [59].

The data from CATA analysis from each judge and each session were gathered and transformed into a unique matrix with attributes coded as (1) detected by a panelist and (0) not detected by the panelist for each descriptor, panelist, and cultivar. The previous matrix was transformed into a contingency table for further analysis.

Cochran’s Q-test (*p* ≤ 0.05) was applied to the raw binary matrix using the ‘RVAideMemoire’ package in R to determine significant differences among cultivars for each sensory attribute [60].

Factorial correspondence analysis (FCA) was performed on the contingency table (with only the fifteen attributes that were significantly different between cultivars according to Cochran’s Q-test) using R (v4.0.1) software to detect clustering and establish relationships between cultivars and sensory attributes [57]. Pearson’s chi squared test (*p* ≤ 0.05) was performed on the contingency table (with only the seven attributes that were significantly different between cultivars according to Cochran’s Q-test) using R software to determine the existence of a statistical relationship between sensory attributes and cultivars [57].

### 4.5. Genetic Diversity Analysis

#### 4.5.1. DNA Extraction

DNA was extracted from 50 mg of leaves of the 43 sweet orange cultivars using a DNeasy^®^ Plant Mini kit (Qiagen, Hilden, Germany) and following the manufacturer’s protocol. The DNA quality and concentration were controlled twice, once using a Nanodrop 2000 spectrophotometer (ThermoFischer Scientific, Waltham, MA, USA), and once by electrophoresis on an agarose gel.

#### 4.5.2. SSR Genotyping

Polymerase chain reaction (PCR) was performed in a 20 µL reaction containing 8 ng of DNA, 0.2 µM of each primer, DreamTaq™ Hot Start PCR Master Mix (ThermoFisher Scientific Waltham, MA, USA) containing *Taq* polymerase, and all other reagents. PCR amplification was performed in a Primus 96+ thermocycler (MWG-Biotech^®^, Luxembourg, Luxembourg), and amplicons were analyzed as described by Luro et al. (2008) with the ten pairs of primers used by Ferrer et al. (2021) [61,62].

## 5. Conclusions

Orange trees show a high phenotypic diversity related to the selection of easily observable traits, such as shape, color, fruit size or skin thickness, succulence, acidity, sugar content, or absence of seeds. These phenotypic variations are the result of spontaneous mutation events occurring in trees in production orchards. Even though the composition of leaf and peel oils is also variable between cultivars, it is less variable than the phenotypic traits. The diversity of PEO aroma profiles between cultivars is sufficient to be differentiated by discriminating tests but insufficient to be easily described by sensory descriptors. Further research is needed to understand the origin of the olfactory differences observed between the zests of different cultivars.

## Figures and Tables

**Figure 1 plants-12-00990-f001:**
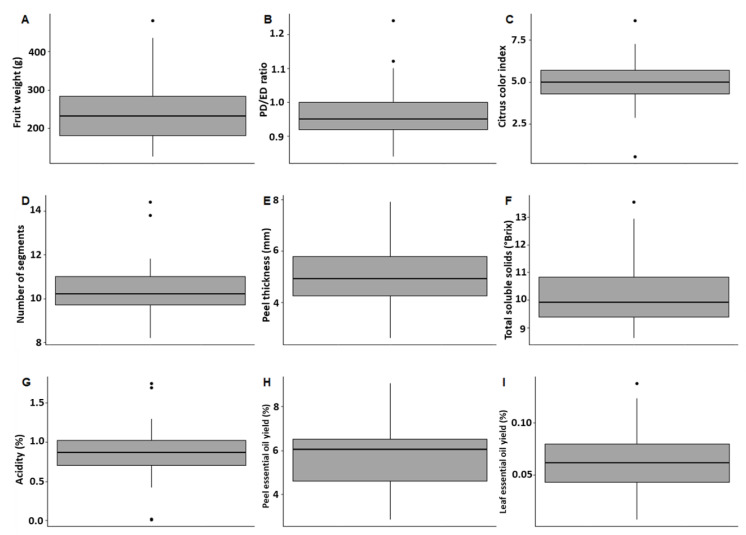
Boxplots representing the distribution of different phenotypic traits among 43 orange cultivars. (**A**) fruit weight, (**B**) fruit shape, (**C**) external fruit peel color, (**D**) number of fruit segments, (**E**) peel thickness, (**F**) total soluble solids, (**G**) titrated acidity, (**H**) yield of PEO, and (**I**) yield of LEO. PD: polar diameter, ED: equatorial diameter.

**Figure 2 plants-12-00990-f002:**
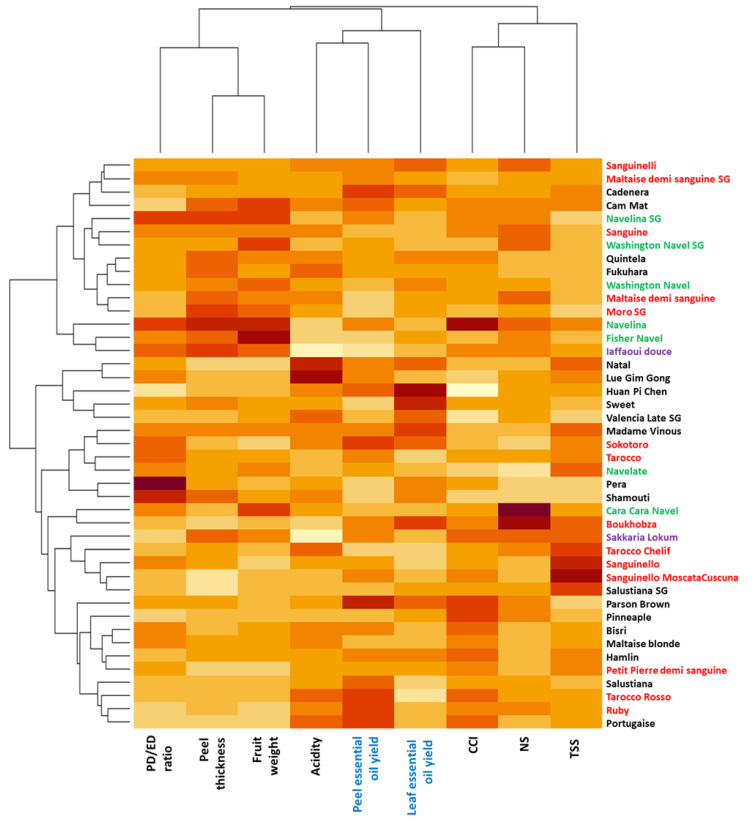
A heatmap representing the phenotypic diversity of the 43 orange cultivars based on the 8 pomological descriptors. The color range from yellow to dark red indicates the degree of value of the observed trait in the orange population, respectively, from weak to strong. Bloody cultivars are represented in red, navels in green, non-acidic in purple, and classic (blond) cultivars in black. PD: polar diameter, ED: equatorial diameter, CCI: citrus color index, NS: number of fruit segment, TSS: total soluble solids.

**Figure 3 plants-12-00990-f003:**
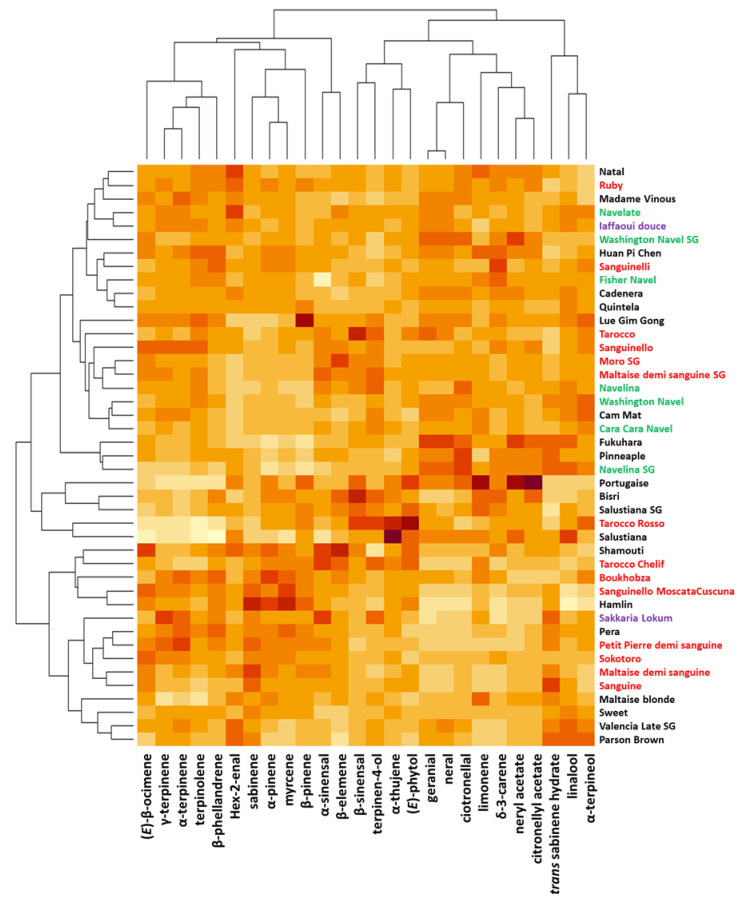
A heatmap representing the diversity of the 43 orange cultivars based on their composition of the 26 major leaf oil compounds. The color range from yellow to dark red indicates the degree of the observed trait value in the orange population. Bloody cultivars are represented in red, navels in green, acid-free in purple, and classic (blond) cultivars in black.

**Figure 4 plants-12-00990-f004:**
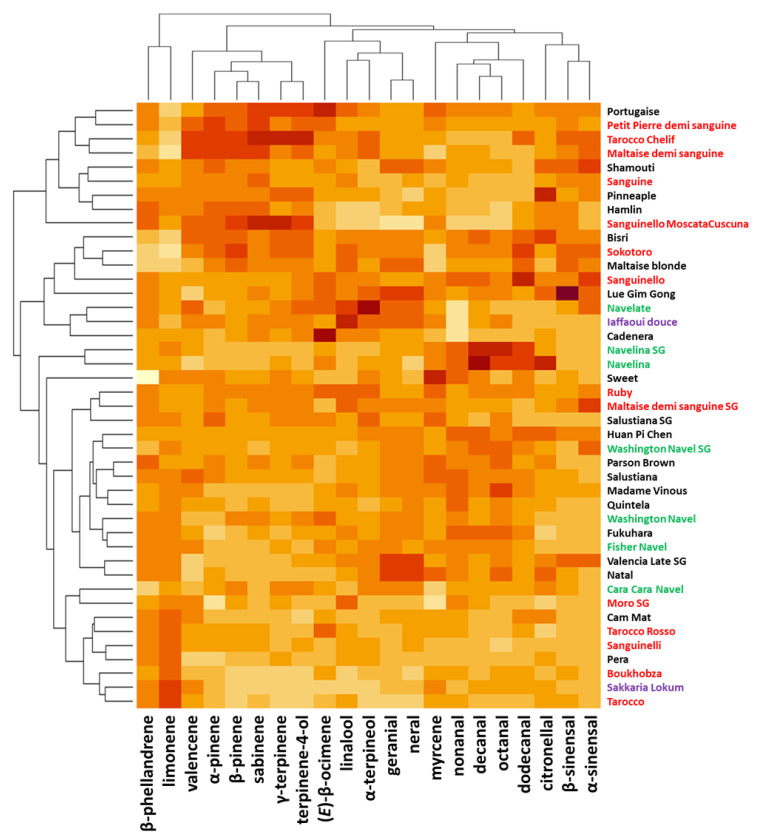
A heatmap representing the diversity of the 43 orange cultivars according to the 20 major peel oil compounds. The color range from yellow to dark red indicates the degree of value of the observed trait in the orange population, respectively, from weak to strong. Bloody cultivars are represented in red, navels in green, acid-free in purple, and classic (blond) cultivars in black.

**Figure 5 plants-12-00990-f005:**
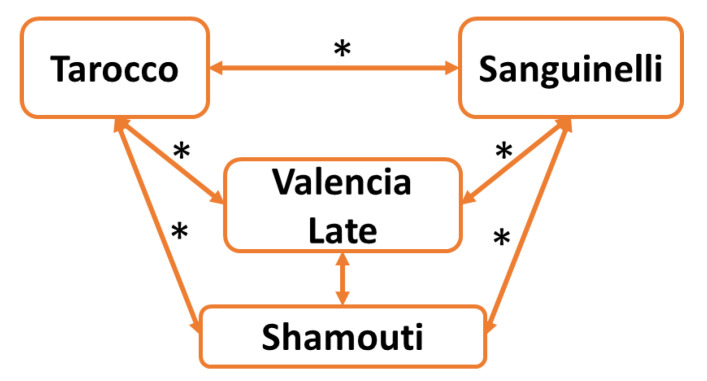
Results of triangle tests conducted on essential oils from the peels of four orange cultivars. The stars above the arrows between two cultivars indicate that a significant difference was observed based on the triangle test (*p* ≤ 0.05).

**Figure 6 plants-12-00990-f006:**
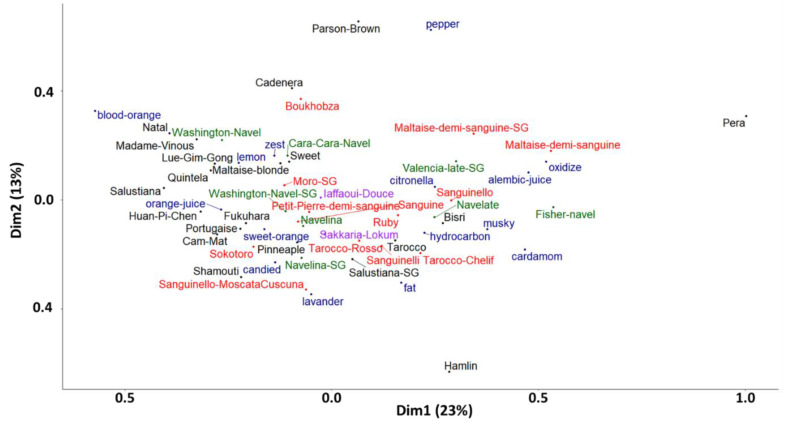
Correspondence Factor Analysis (CFA) representing the diversity of the 43 orange cultivars according to the 15 descriptors (represented in blue) of the aroma profiles of the essential oils is significantly different between cultivars according to the Cochran test (*p* ≤ 0.05). ‘Blood’ cultivars are represented in red, ‘navel’ in green, ‘acidless’ in purple, and ‘blond’ cultivars in black.

**Table 1 plants-12-00990-t001:** Proportion of oxygenated compounds in orange leaf oil for the three types of cultivars. The first number is the mean value, the second is the standard deviation, and the letter is the statistical group according to Tukey’s test (*p* ≤ 0.05).

Cultivars	Blond	Blood	Navel
Monoterpene aldehydes	5.51 ± 2.48 ab	3.72 ± 1.49 b	7.35 ± 1.94 a
Monoterpene alcohols	20.94 ± 4.72 a	20.43 ± 4.52 a	23.35 ± 3.49 a
Monoterpene esters	0.69 ± 0.45 a	0.42 ± 0.18 a	0.76 ± 0.30 a
Oxygenated sesquiterpenes	2.60 ± 0.76 a	2.84 ± 0.87 a	2.43 ± 0.40 a
Aliphatic aldehydes	0.75 ± 0.22 a	0.57 ± 0.14 a	0.64 ± 0.30 a

**Table 2 plants-12-00990-t002:** Proportion of oxygenated compounds in orange peel oil for the three typical cultivar types. The first number is the mean value, the second is the standard deviation, and the letter is the statistical group according to Tukey’s test (*p* ≤ 0.05).

Cultivars	Blond	Blood	Navel
Monoterpene aldehydes	0.15 ± 0.05 a	0.10 ± 0.04 b	0.15 ± 0.03 ab
Monoterpene alcohols	0.92 ± 0.30 a	0.96 ± 0.32 a	0.77 ± 0.37 a
Oxygenated sesquiterpenes	0.03 ± 0.04 a	0.04 ± 0.03 a	0.01 ± 0.01 a
Aliphatic aldehydes	0.54 ± 0.16 ab	0.41 ± 0.15 b	0.70 ± 0.30 a

## Data Availability

All the data produced and used in this article are available in the manuscript or in the Appendix A.

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
