# Peer review of "Correspondence between the Compositional and Aromatic Diversity of Leaf and Fruit Essential Oils and the Pomological Diversity of 43 Sweet Oranges (Citrus x aurantium var sinensis L.)"

_plants, 2023, doi:10.3390/plants12050990_

Round 1

Reviewer 1 Report

The article is interesting, based on a large quantity of experimental  data  and well presented . The results are interesting for the users of orange derived essential oils as mentioned in the article and exemplified for Cointreau producers.

 There is a large degree of similarity ( 29 %, as resulted from a Turnitin survey) in the description of the section Material and methods with a related article  of the group ( ref 37) regarding the sour varieties of Oranges ( 24% similarity). Taking into account that is difficult to describe in different words analysis procedures I think that althouth similarity may be explained,  those phrases ( highlighted in red in the Turnitin report ) may be edited in order to decrease this similarity degree.

Author Response

The article is interesting, based on a large quantity of experimental data and well presented. The results are interesting for the users of orange derived essential oils as mentioned in the article and exemplified for Cointreau producers.

Authors: We thank the reviewer for this evaluation and appreciation

 There is a large degree of similarity (29 %, as resulted from a Turnitin survey) in the description of the section Material and methods with a related article of the group (ref 37) regarding the sour varieties of Oranges (24% similarity). Taking into account that is difficult to describe in different words analysis procedures I think that although similarity may be explained, those phrases (highlighted in red in the Turnitin report) may be edited in order to decrease this similarity degree.

Authors: It is a work of thesis, which was, carried out under the same experimental conditions on two species (more than forty cultivars for each one) the sweet oranges (C. sinensis) and the bitter oranges (C. aurantium), in the same laboratories and by the same authors. Since we are the authors of both articles, we cannot be accused of plagiarism. If nevertheless you consider that it is essential to reduce the degree of similarity of the "Materials and Methods" section, we propose to delete the whole analytical procedure and to put instead the reference of our previous publication (Ferrer, V.; Costantino, G.; Paoli, M.; Paymal, N.; Quinton, C.; Ollitrault, P.; Tomi, F.; Luro, F. Intercultivar Diversity of Sour Orange (Citrus aurantium L.) Based on Genetic Markers, Phenotypic Characteristics, Aromatic Compounds and Sensorial Analysis. Agronomy, 2021, 11 (6), 1084. https://doi.org/10.3390/agronomy11061084) ?

Reviewer 2 Report

Manuscript Number: plants-2120619, titled:

Correspondence between the compositional and aromatic diversity of leaf and fruit essential oils and pomological diversity of 43 sweet oranges (Citrus x aurantium var sinensis L.).

Review 1 – 17 December 2022

Dear Editor of Plants

the argument is interesting and well treated in the large part of the manuscript. The introduction section has to be improved and well argued. There is some inaccuracy in the text.

I suggest a major revision

To the Authors (in detail):

1)    the argument is interesting and well treated in the large part of the manuscript. The introduction section has to be improved and well argued. There is some inaccuracy in the text.

2)    The manuscript has to be re-arranged by using the instructions for authors of Plants, for example, the section Materials and Methods has to be included after the discussion section. Please, verify and re-arrange the whole manuscript. Some inaccuracy in the text and tables.

3)    Introduction section, please, explain that the PEO composition of a citrus fruit is influenced by many factors such as: the environment [X1], the harvest year [X2], the harvest date [X3]; the extraction system [X4]. Support this statement with proper references and do not cumulate the bibliography at the end of the sentence but include the number after each influencing factor. I suggest to find, read and discuss:

[X1] Effect of Environmental Conditions on the Yield of Peel and Composition of Essential Oils from Citrus Cultivated in Bahia (Brazil) and Corsica (France).

Agronomy 2020, 10, 1256; doi:10.3390/agronomy10091256

[X2] The peel essential oil composition of bergamot fruit (Citrus bergamia, Risso) of Reggio Calabria  (Italy): a review.

Emirates Journal of Food and Agriculture  32 (11) 835-845 (2020)

doi: 10.9755/ejfa.2020.v32.i11.2197

[X3] Changes of Peel Essential Oil Composition of Citrusaurantium L. During Fruit Maturation in Iran, Journal of Essential Oil Bearing Plants, 18:4, 1006-1012 (2015),

 DOI: 10.1080/0972060X.2014.977564

[X4] Cold Pressing, Hydrodistillation and Microwave Dry Distillation of Citrus Essential Oil from Algeria: A Comparative Study.

Electronic Journal of Biology, 2016, Vol.S1: 30-41

4)    Sub-section 2.1, please, include the fertilizers used (type, quantity and period); include the irrigation (type, quantity, period);

5)    Sub-section 2.1, please, indicate the minimum and maximum temperature during the year and the difference between day and night;

6)    2.2.1 sub-section, please, indicate the day after blossom for collection of fruits;

7)    Sub-section 2.2.2., indicate the significance value;

8)    Sub-section 2.3.2, line 141 and in the whole manuscript, tables and figures, when you indicate a temperature, separate the numeric value by the symbol: 50 °C and not 50°C;

9)    Sub-section 2.3.3, lines 147-148, please, verify carefully the GC column data, in addition, include the word “length” for consistency with other 2 parameters;

10) Line 158, include “length” after 50 m;

11) Sub-section 2.4.1. , indicate if panellists were smokers or not;

12) 2.4.1. sub-section, please evidence if panellists were trained or not;

13) Sub-section 3.2, line 233-234, verify the font;

14) Sub-section 3.2 and in the whole manuscript, sometime you use cultivar and sometime variety. These two words are not synonyms, please, verify and re-arrange the manuscript;

15) Table 1: small or capital letters for the head of the columns?

16)  Line 295, separate the text by the table 1;

17) Line 340, separate the text by the table 2;

18) Please, write in blue color or evidence differently the corrections you will do.

I suggest a major revision

Regards.

Author Response

Authors: We thank reviewer for his recommendations and suggestions to improve our manuscript. Our answers are written in blue

To the Authors (in detail):

1)    the argument is interesting and well treated in the large part of the manuscript. The introduction section has to be improved and well argued. There is some inaccuracy in the text.

Authors: See responses to Note 3).

2)    The manuscript has to be re-arranged by using the instructions for authors of Plants, for example, the section Materials and Methods has to be included after the discussion section. Please, verify and re-arrange the whole manuscript. Some inaccuracy in the text and tables.

Authors: The order of the sections has been changed to reflect the recommendations for authors and the sections and references have been renumbered accordingly. 

3)    Introduction section, please, explain that the PEO composition of a citrus fruit is influenced by many factors such as: the environment [X1], the harvest year [X2], the harvest date [X3]; the extraction system [X4]. Support this statement with proper references and do not cumulate the bibliography at the end of the sentence but include the number after each influencing factor. I suggest to find, read and discuss:

Authors: As suggested by the reviewer, we added in the introduction section the following information about the factors that can modify the citrus essential oil composition:

Nevertheless, the PEO composition of a citrus fruit can be influenced by many factors [15] such as: the environment [16], the harvest year [17], the harvest date [18, 19]; the rootstock [20, 21], the extraction system [22, 23], the storage conditions [24], and the health of the tree [25-27]. Some examples of the effects of these different factors of variation in citrus PEO composition are presented hereafter. The environments of Bahia (Brazil) and Corsica (France) modified the proportion of limonene, neral, geranial and derivatives in citron and lemon PEOs [16]. Gioffrè, et al. reported the chemical composition of the bergamot peel essential oil described in the studies carried out from the 1984-1985 until the 2018-19 and highlighted noticeable variations: the major component limonene ranging between 31.10% and 44.48% [17]. The PEO composition was richer in oxygenated compounds at early fruit development stages, with an aromatic profile presenting greener notes than in mature orange fruit [19]. The rootstock influenced the yield and a little bit the composition of the orange PEO, but with a very low impact on flavor [20]. The citrus EO extracted by microwave - Clevenger method, contained elevated quantities of oxygenated compounds comparatively to hydro-distillation and cold pressing methods [22]. In sour orange, linalyl acetate and linalool drastically decrease when citrus are ripening [10]. The longer the oil was stored, the more important the changes in the composition were the following: limonene, myrcene, γ-terpinene and linalyl acetate percentages decreased when the temperature and the time of storage increased [24]. The PEOs of orange trees infected by huanglongbing (HLB) had lower concentrations of valencene, octanal, and decanal, and were abundant in oxidative/dehydrogenated terpenes, such as carvone and limonene oxides [27].

The additional references:

15      Gaff, M.; Esteban‐Decloux, M.; Giampaoli, P. Bitter Orange Peel Essential Oil: A Review of the Different Factors and Chemical Reactions Influencing Its Composition. Flavour Fragr. J. 2020, 35, 247–269, doi:10.1002/ffj.3570. 

16      Luro, F., Garcia Neves, C., Costantino, G., da Silva Gesteira, A., Paoli, M., Ollitrault, P., Tomi, F., Micheli, F. and Gibernau, M. Effect of Environmental Conditions on the Yield of Peel and Composition of Essential Oils from Citrus Cultivated in Bahia (Brazil) and Corsica (France). Agronomy, 2020, 10, 1256, https://doi.org/10.3390/agronomy10091256

17      Gioffrè, G., Ursino, D., Labate, M. L. C., and Giuffrè, A. M. The Peel Essential Oil Composition of Bergamot Fruit (Citrus Bergamia, Risso) of Reggio Calabria (Italy): A Review.” Emirates Journal of Food and Agriculture, 2020, 32, 835-845, https://doi.org/10.9755/ejfa.2020.v32.i11.2197.

18      Rowshan, V., and Najafian, S. Changes of Peel Essential Oil Composition of Citrus aurantium L. During Fruit Maturation in Iran, Journal of Essential Oil Bearing Plants, 2015, 18:4, 1006-1012, https://doi.org/10.1080/0972060X.2014.977564

19      Ferrer, V., Paymal, N., Quinton, C., Tomi F., and Luro, F. Investigations of the Chemical Composition and Aromatic Properties of Peel Essential Oils throughout the Complete Phase of Fruit Development for Two Cultivars of Sweet Orange (Citrus sinensis (L.) Osb.). Plants, 2022, 11, 2747, https://doi.org/10.3390/plants11202747

20      Ferrer, V., Paymal, N., Quinton, C., Costantino, G., Paoli, M., Froelicher, Y., Ollitrault, P., Tomi, F., Luro, F. Influence of the Rootstock and the Ploidy Level of the Scion and the Rootstock on Sweet Orange (Citrus sinensis) Peel Essential Oil Yield, Composition and Aromatic Properties. Agriculture, 2022, 12, 214. https://doi.org/10.3390/agriculture12020214.

21      Benjamin, G.; Tietel, Z.; Porat, R. Effects of Rootstock/Scion Combinations on the Flavor of Citrus Fruit. J. Agric. Food Chem. 2013, 61, 11286–11294, https://doi.org/10.1021/jf402892p

22      Ferhat, M.A., Boukhatem, M.N., Hazzit M, Meklati, B.Y., Chemat, F. Cold Pressing, Hydrodistillation and Microwave Dry Distillation of Citrus Essential Oil from Algeria: A Comparative Study. Electronic J. Biol. 2016, 1, 30-41

23      Farahmandfar, R.; Tirgarian, B.; Dehghan, B.; Nemati, A. Changes in Chemical Composition and Biological Activity of Essential Oil from Thomson Navel Orange ( Citrus Sinensis L. Osbeck) Peel under Freezing, Convective, Vacuum, and Microwave Drying Methods. Food Sci. Nutr. 2020, 8, 124–138, https://doi.org/10.1002/fsn3.1279

24      Usai, Marianna.; Arras, Giovannni.; Fronteddu, Franco. Effects of Cold Storage on Essential Oils of Peel of Thompson Navel Oranges. J. Agric. Food Chem. 1992, 40, 271–275, https://doi.org/10.1021/jf00014a021.

25      Zouaghi, G.; Najar, A.; Aydi, A.; Claumann, C.A.; Zibetti, A.W.; Ben Mahmoud, K.; Jemmali, A.; Bleton, J.; Moussa, F.; Abderrabba, M.; et al. Essential Oil Components of Citrus Cultivar ‘Maltaise Demi Sanguine’ ( Citrus Sinensis ) as Affected by the Effects of Rootstocks and Viroid Infection. Int. J. Food Prop. 2019, 22, 438–448, https://doi.org/10.1080/10942912.2019.1588296

26      Xu, B.M.; Baker, G.L.; Sarnoski, P.J.; Goodrich-Schneider, R.M. A Comparison of the Volatile Components of Cold Pressed Hamlin and Valencia (Citrus Sinensis (L.) Osbeck) Orange Oils Affected by Huanglongbing. J. Food Qual. 2017, 2017, 1–20, https://doi.org/10.1155/2017/6793986

27      Sun, X.; Yang, H.; Zhao, W.; Bourcier, E.; Baldwin, E.A.; Plotto, A.; Irey, M.; Bai, J. Huanglongbing and Foliar Spray Programs Affect the Chemical Profile of “Valencia” Orange Peel Oil. Front. Plant Sci. 2021, 12, 611449, https://doi.org/10.3389/fpls.2021.611449.

4)    Sub-section 2.1, please, include the fertilizers used (type, quantity and period); include the irrigation (type, quantity, period);

Authors: This information was added as following:

The planting density is 416 trees per hectare. Fertilization was provided in 3 annual applications (February, May and August) for a total of 450 kg per hectare with Multigro fertilizer (Haifa, France), 15% nitrogen, 23% phosphoric anhydride, 14% potassium oxide and 2% magnesium oxide. The trees were irrigated by micro-sprinklers, according the estimated needs of the tree with respect to the potential evapotranspiration (Etp), rainfall (P) and cultural coefficient (Kc): (Etp–P) x Kc. The water supply during the dry summer period was approximately, input of 25 mm per week for a total of about 1000 L per tree, per year.

5)    Sub-section 2.1, please, indicate the minimum and maximum temperature during the year and the difference between day and night;

Authors: This information was added as following:

The average annual temperature of the year from blossom (May 2018) to leaf picking (June 2019) was 15.2 °C with a mean maximum of 30.3 °C in July and a mean lower of 3.6 °C in January. The mean temperature of day and night difference during this period was 11.7 ± 2.9 °C.

6)    2.2.1 sub-section, please, indicate the day after blossom for collection of fruits;

Authors: This information was added.

7)    Sub-section 2.2.2., indicate the significance value;

Authors: This information (p ≤ 0.05) was added.

8)    Sub-section 2.3.2, line 141 and in the whole manuscript, tables and figures, when you indicate a temperature, separate the numeric value by the symbol: 50 °C and not 50°C;

Authors: We have checked the whole manuscript and made the modifications

9)    Sub-section 2.3.3, lines 147-148, please, verify carefully the GC column data, in addition, include the word “length” for consistency with other 2 parameters;

Authors: We modified the text: 0.22 mm instead of 22 mm

10) Line 158, include “length” after 50 m;

Authors: Done

11) Sub-section 2.4.1. , indicate if panellists were smokers or not;

Authors: See answer of remark 12).

12) 2.4.1. sub-section, please evidence if panellists were trained or not;

Authors: We added the information requested in remarks 11) and 12) as following:

The panelists (ten women and eight men aged between 22 and 55) are not smoker people from the Cointreau laboratory trained to control the quality of the liquors during the manufacturing process and participate in daily tastings.

13) Sub-section 3.2, line 233-234, verify the font;

Authors: We have checked the font of whole manuscript and made the modifications

14) Sub-section 3.2 and in the whole manuscript, sometime you use cultivar and sometime variety. These two words are not synonyms, please, verify and re-arrange the manuscript;

Authors: Indeed, the term "variety" is inappropriate for our studied plant material and has therefore been replaced throughout the document by "cultivar"

15) Table 1: small or capital letters for the head of the columns?

Authors: The correction was done.

16)  Line 295, separate the text by the table 1;

Authors: Done.

17) Line 340, separate the text by the table 2;

Authors: Done.

18) Please, write in blue color or evidence differently the corrections you will do.

Authors: The answer to the remarks is written in blue but in the text, the corrections are visible with the follow-up function of text modification.

Round 2

Reviewer 2 Report

Manuscript Number: plants-2120619, titled:

 Correspondence between the compositional and aromatic diversity of leaf and fruit essential oils and pomological diversity of 43 sweet oranges (Citrus x aurantium var sinensis L.).

Review 2 – 19 February 2023

Dear Editor of Plants

the argument is interesting and well treated. The authors have included all my comments.

I suggest the publication in the current form.

Regards.